# Acute and Delayed Effects of Mechanical Injury on Calcium Homeostasis and Mitochondrial Potential of Primary Neuroglial Cell Culture: Potential Causal Contributions to Post-Traumatic Syndrome

**DOI:** 10.3390/ijms23073858

**Published:** 2022-03-31

**Authors:** Zanda Bakaeva, Mikhail Goncharov, Irina Krasilnikova, Arina Zgodova, Daniil Frolov, Ekaterina Grebenik, Peter Timashev, Vsevolod Pinelis, Alexander Surin

**Affiliations:** 1Laboratory of Neurobiology and Fundamentals of Brain Development, National Medical Research Center of Children’s Health, 119991 Moscow, Russia; irinakrsl81@gmail.com (I.K.); zgodova.ae@nczd.ru (A.Z.); pinelis@mail.ru (V.P.); surin_am@mail.ru (A.S.); 2Department of General Biology and Physiology, Kalmyk State University Named after B.B. Gorodovikov, 358000 Elista, Russia; 3Center of Life Sciences, Skolkovo Institute of Science and Technology, 121205 Moscow, Russia; mikhail.goncharov@uksh.de; 4International School “Medicine of the Future”, Sechenov First Moscow State Medical University (Sechenov University), 119146 Moscow, Russia; 5Institute of Information Technologies, MIREA-Russian Technological University, 119454 Moscow, Russia; d4n.fro1@gmail.com; 6Institute for Regenerative Medicine, Sechenov First Moscow State Medical University (Sechenov University), 119146 Moscow, Russia; grebeneka@gmail.com (E.G.); timashev.peter@gmail.com (P.T.); 7World-Class Research Center “Digital Biodesign and Personalized Healthcare”, Sechenov First Moscow State Medical University (Sechenov University), 119146 Moscow, Russia; 8Semenov Institute of Chemical Physics, Russian Academy of Sciences, 119334 Moscow, Russia; 9Chemistry Department, Lomonosov Moscow State University, 119991 Moscow, Russia; 10Institute of General Pathology and Pathophysiology, Russian Academy of Medical Sciences, 125315 Moscow, Russia

**Keywords:** mechanical trauma, post-traumatic syndrome, neuroreparation, neuroglial culture, GFAP, β-III tubulin, [Ca^2+^]_i_, ΔΨm, NMDAR

## Abstract

In vitro models of traumatic brain injury (TBI) help to elucidate the pathological mechanisms responsible for cell dysfunction and death. To simulate in vitro the mechanical brain trauma, primary neuroglial cultures were scratched during different periods of network formation. Fluorescence microscopy was used to measure changes in intracellular free Ca^2+^ concentration ([Ca^2+^]_i_) and mitochondrial potential (ΔΨm) a few minutes later and on days 3 and 7 after scratching. An increase in [Ca^2+^]_i_ and a decrease in ΔΨm were observed ~10 s after the injury in cells located no further than 150–200 µm from the scratch border. Ca^2+^ entry into cells during mechanical damage of the primary neuroglial culture occurred predominantly through the NMDA-type glutamate ionotropic channels. MK801, an inhibitor of this type of glutamate receptor, prevented an acute increase in [Ca^2+^]_i_ in 99% of neurons. Pathological changes in calcium homeostasis persisted in the primary neuroglial culture for one week after injury. Active cell migration in the scratch area occurred on day 11 after neurotrauma and was accompanied by a decrease in the ratio of live to dead cells in the areas adjacent to the injury. Immunohistochemical staining of glial fibrillary acidic protein and β-III tubulin showed that neuronal cells migrated to the injured area earlier than glial cells, but their repair potential was insufficient for survival. Mitochondrial Ca^2+^ overload and a drop in ΔΨm may cause delayed neuronal death and thus play a key role in the development of the post-traumatic syndrome. Preventing prolonged ΔΨm depolarization may be a promising therapeutic approach to improve neuronal survival after traumatic brain injury.

## 1. Introduction

Pathological mechanisms underlying the detrimental effects of traumatic brain injury (TBI) are divided into primary and secondary injury mechanisms. Primary damage is due to the mechanical impact itself (force, degree of penetration, and area of damage) and develops in the first few seconds after the trauma. The most important aspect of this stage is necrotic cell death in the area of the injured brain tissue. Secondary damage is longer-term (hours-days) and is mediated by primary damage mechanisms [1]. They include pathological depolarization due to shear stress, changes in cerebral blood flow (CBF) dynamics, as well as neuro-inflammatory processes [2] caused by glial and immune cells coming from the bloodstream during blood–brain barrier (BBB) permeabilization.

At the same time, the central mechanism of long-term neuronal death induced by TBI is glutamate excitotoxicity [3]. Briefly, the exposure of neural tissue to kinetic energy leads to the immediate depolarization of neurons, causing an abundant release of excitatory neurotransmitters, the most prevalent of which is glutamate [4,5]. It was previously shown that concussion leads to a 50-fold local increase in the concentration of extracellular glutamate in the human brain [6]. In addition to mechanically depolarized or structurally impaired neurons, nutritional and oxygen deprivation also contributes to this dramatic increase in glutamate concentration, leading to an inversion of astrocyte glutamate transporter [7] and impaired glutamate uptake by presynaptic neurons [8].

Subsequently, glutamate causes Ca^2+^-dependent neurotoxicity due to prolonged activation of NMDA-type ionotropic glutamate receptors, which have a high permeability to extracellular Ca^2+^ and Na^+^. A sharp increase in intracellular free Ca^2+^ concentration ([Ca^2+^]_i_) activates various mechanisms of cell death, leading to additional loss of neurons long after the primary damage. Therefore, a decrease in [Ca^2+^]_i_ after mechanical damage to neurons in vitro correlates with a decrease in neuronal damage and an increase in their viability. A reproduction of such results in general indicates the neuroprotective activity of the studied preparations [9].

The increase in intracellular Na concentration near the NMDA channel also plays a substantial role in the pathophysiology of TBI. This leads to a massive membrane depolarization and increase in H_2_O retention, direct water movement into the intracellular compartment, and, consequently, cytotoxic edema. At the same time, Na^+^-dependent depolarization causes Cl^−^ influx, which only aggravates neuronal swelling and, consequently, cell death [10]. Another factor increasing intracellular Na^+^ and Cl^−^ is NKCC1, which has been shown [11] to be upregulated after TBI. The main consequence of cytotoxic edema, cell death, was previously attributed to its positive effect on intracellular Ca^2+^ accumulation [12]. However, another study [10] suggested that cell death in neuronal edema occurs even in the absence of dramatic [Ca^2+^] changes, and can be prevented by blocking depolarization-dependent Cl^-^ influx.

There are various in vitro models of mechanical neurotrauma. Models of acute penetrating impact allow the reproduction of the process of neuronal damage using a flexible plastic pipette tip [13], a circular slicer, or a razor [14,15]. These and other models [16,17] suggest an acute disruption of cell integrity, exposing neighboring neurons to excitotoxic effects and causing glial cells activation. A limitation of all these models is the lack of standardized parameters for the dynamics of mechanical injury and assessment of its severity, primarily based on the number of damaged cells, which introduces great uncertainty in the interpretation of results.

This work presents a simple and easily reproducible model of mechanical neurotrauma for in vitro studies of primary and secondary neuronal damage. This model allowed us to study the acute (seconds, minutes) and delayed (7 days after mechanical impact) effects of the scratch on cell viability, the dynamics of [Ca^2+^]_i_, and mitochondrial potential (ΔΨm) changes in neurons. Studying these parameters, we traced the dynamics of the scratch healing and determined which neuroglial culture cells were the first to migrate to the zone of mechanical damage. Thus, we substantiated a new in vitro model of TBI, which can be further used to study the neuroprotective compounds we previously tested in an experimental model of glutamate excitotoxicity [18].

## 2. Results

### 2.1. Effects of Mechanical Trauma on Cell Survival and Regeneration in Neuroglial Culture

Scratch injury caused the immediate death of all cells in the damaged area (Figure 1, Syto-13, Hoechst33342). Usually, some cell debris was observed in the scratched zones (Figure 1, EthD-1). These zones were still clearly visible three days (at 7th DIV) after disruption of the neuronal network (Figure 1, Merge). Morphological analysis showed a decrease in the ratio of live/dead cells in the areas located nearby the scratch (Figure 2A). The spatial organization of cells was changing. They grouped around predominantly dead cells (Figure 1, EthD-1, Merge). It is important to note that the number of living cells increased significantly in the scratch area on the 11th and 14th days after the injury (14 and 17 DIV, respectively) (Figure 2B). The penetration and growth of neurites, as well as diffusion of neurons into the zone of the scratch until its almost complete recovery, was observed (Figure 3B). The ratio of live to dead cells outside the scratched area gradually decreased until 17th DIV (Figure 2A). This effect may be related, at least partially, to the primary cell culture maturation. Many studies used a primary neuronal cortical culture no older than 9–15 DIV [19,20,21,22,23,24], which were different from the long-term neural cultures differentiated from induced pluripotent stem cells [25,26].

Interestingly, the decrease in the ratio of living/dead cells in the areas near the scratch on days 11 and 14 after the injury was synchronized with the increase in the number of living cells in the damaged area on the same days (Figure 2A,B). Such a gradual decrease in the proportion of living cells in areas near scratches may be attributed to the migration of cells into the damaged area (Figure 3A,B). It is known that neurons can express chemokine receptors, which allows directional migration of poorly differentiated neural cells during normal brain development [27].

Immunohistochemical staining for GFAP and β-III tubulin showed that cells that appeared earlier than others in the scratch area were neurons (Figure 3A and Figure 4A). We found characteristic green stripes remaining after the destruction of astrocytes in the damaged area. This did not prevent the neurons from migration to the damaged area, but supposedly attracted them (Figure 3B). Figure 3C shows that, up to the 10th day of cultivation, there were a few astrocytes in the mixed neuroglial intact culture. Basically, these were protoplasmic astrocytes with numerous short-branched processes. They contained less fibrous material. A low expression of glial fibrillar protein was observed. These cells surrounded the soma of neurons and their dendrites and synaptic contacts.

Figure 4 shows the dynamics of the scratched area overgrowth. On the 3rd day after scratching (7 DIV), the first neurons, but no astrocytes, appeared in the damaged area, and a network of a few neurites (Figure 4A) was visible. On the 7th day after scratching (10 DIV), the number of migrated neurons and their processes increased, followed by the appearance of astrocytes in the damaged area (Figure 4B). On the 10th day after scratching (14 DIV), the morphology of astrocytes changed: their bodies increased, and there were many cells with a large nucleus and numerous branched processes (Figure 4C). A high expression of glial fibrillar protein (Figure 4C) was observed as an intact cell culture on 14 DIV (Figure 4D). Oddly, there were a few neurons (Figure 4C) in the damaged area, which may be explained by the depletion of the regenerative potential of the primary neuroglial culture on 14 DIV.

### 2.2. Effect of Mechanical Injury on the Parameters of Acute Changes in [Ca^2+^]_i_ and ΔΨm

Previously, we observed different dynamics of changes in neuronal [Ca^2+^]_i_ in response to Glu [18,28]. The standard variability in cellular responses to glutamate is shown in Figure 5. For instance, cell “1” marked with a red circle (Figure 5B) had a short duration of lag-phase (Figure 5A) before DCD development and did not recover initial [Ca^2+^]_i_ during the period of incubation in Ca^2+^-free media in experimental stage (e) (Figure 5C). Cell “2”, marked with a green circle (Figure 5B), had an almost 5 min long lag-phase in experimental stage (c) (Figure 5A) and almost completely restored the initial [Ca^2+^]_i_ (Figure 5C). These drastic differences in the dynamics of [Ca^2+^]_i_ alterations may be attributed to the differential pattern of NMDA receptors expression on the cytoplasmic membrane of those two neurons, and therefore different sensitivity to Glu and the ability to maintain calcium homeostasis. Astrocytes, which are also present in the neuroglial culture, can respond to glutamatergic stimulation as well, as they express NMDA and AMPA receptors [29], but the kinetics and amplitude of [Ca^2+^]_i_ changes are much milder in these cells, similar to cell “3.” This could be due to the lower density and different isoform of NMDA receptors expressed in astrocytes, and also due to the fact that these cells store much more substantial amounts of glycogen in comparison to neurons, therefore having a different metabolic capacity.

Fluorescent microscopy images of the primary rat cortical cell culture site before scratching and 10 s after scratching are shown in Figure 6A,B. It can be seen that the [Ca^2+^]_i_ rapid increase caused by the scratch was higher for those cells that were located no further than 100 microns from the scratch (Figure 6B,D). Graphs of ΔΨm change demonstrate the synchronism of the growth of [Ca^2+^]_i_ and mitochondrial depolarization (Figure 6D,G). A similar synchronism of [Ca^2+^]_i_ and ΔΨm was repeatedly observed under the action of toxic doses of glutamate on cultured neurons [18,28,30]. The fluorescent microscopy images and the graphs of [Ca^2+^]_i_ and ΔΨm changes after the addition of glutamate are shown in Figure 6C,F,I, respectively.

We checked whether the mechanical damage-dependent rapid increase in [Ca^2+^]_i_ and drop in ΔΨm can be caused by the Ca^2+^ influx through NMDA channels. For this, the application of scratches was carried out in a buffer containing MK-801, a noncompetitive inhibitor of NMDA receptors. In the presence of an inhibitor (10 μM), the fraction of cells with a strong increase in [Ca^2+^]_i_ and a significant drop in ΔΨm decreased 8.5 ± 1.5 times (4 experiments, 111 cells; Figure 6E,H). However, part of the cells still underwent a sharp increase in [Ca^2+^]_i_ despite inhibition of the NMDA receptors. This effect could not be assigned to the incomplete blockade of NMDA receptors by MK-801, as the subsequent addition of Glu did not cause a jump in [Ca^2+^]_i_ and a strong fall in ΔΨm in no cell (Figure 6E,H). This is fully consistent with the notion that NMDA channels are the dominant paths of Ca^2+^ entry into the cell [31]. Obviously, in culture, there were cells in which there were highly conductive paths of Ca^2+^ entry into the cytoplasm in addition to the NMDA channels and channels activated by the entrance of Ca^2+^ and Na^+^ through Glu-controlled channels. This assumption is confirmed by the fact that in the buffer devoid of Ca^2+^, the changes in [Ca^2+^]_i_ and ΔΨm were completely absent (not shown).

### 2.3. Delayed Changes in [Ca^2+^]_i_ and ΔΨm in Cells of Mechanically Injured Neuroglial Cultures under Conditions of Glutamate Excitotoxicity

To assess the delayed effect of mechanical trauma on calcium homeostasis of cortical neurons, two consecutive series of experiments were performed on mixed primary neuroglial sister cultures on days 3 and 7 after scratching of culture (7 and 11 DIV, respectively). To assess the functional status of intracellular calcium homeostasis systems of neurons, artificial conditions of glutamate excitotoxicity were created using a subtoxic dose of glutamate (33 μM).

In the first series of experiments (7 DIV), no significant differences were found between the groups in the parameters of [Ca^2+^]_i_ and ΔΨm dynamics in response to the Glu; the lag-period of the development of delayed calcium dysregulation (DCD) in the damaged culture did not differ from the control (Figure 7A).

In the second series of experiments (11 DIV), it was shown that compared to neurons from control cultures that were not mechanically injured, neurons from injured cultures had impaired calcium homeostasis (Figure 7B). Cells from injured cultures showed a more rapid development of DCD in response to Glu administration than the cells from intact cultures (Figure 8A_1,2_).

Important parameters reflecting the dynamics of DCD are the area under the Glu and EGTA curves (Figure 8B_1_,C_1_). These parameters reflect, respectively, the process of Ca^2+^ entry into the cytoplasm during the activation of glutamate receptors and, conversely, a decrease in its cytoplasmic concentration during the energy-dependent removal of Ca^2+^ carried out by plasma and mitochondrial transporters.

The integrated fluorescent response reflects the change in fluorescence intensity of Fura-FF in cells during the 15 min action of Glu. It was significantly higher in the neurons from the injured cultures than in the intact ones (*p* < 0.01) (Figure 8B_2_). A similar result was observed during the recovery period of calcium homeostasis after exposure to Glu in EGTA solution (*p* < 0.01) (Figure 8C_2_). In addition, neurons from mechanically disintegrated cultures poorly restored [Ca^2+^]_i_ to initially lower values than control cells (*p* < 0.01) (Figure 8D_1,2_). Consequently, higher rates of [Ca^2+^]_i_ against EGTA in cells from injured cultures may indicate that the activity of transport systems that pump calcium from the neuronal cytoplasm was deteriorated.

In order to assess the contribution of mitochondria to the calcium-dependent processes occurring during the experiment, in parallel with the assessment of the dynamics of [Ca^2+^]_i_, we measured the mitochondrial potential (ΔΨm).

The recovery rate of ΔΨm during the “washout” period (administration of EGTA) was estimated by the slope of the graph reflecting the change in Rh123 (Figure 8E_1_). The rate of mitochondrial potential recovery in neurons of injured cultures was significantly lower than in neurons of intact cultures (Figure 8E_2_).

There were similar changes in the integrated fluorescent response both for Rh123 and Fura-FF during the application of EGTA solution. This fact not only indicates that the [Ca^2+^]_i_ and ΔΨm changes were synchronous, but also confirms the hypothesis of the participation of mitochondria in glutamate-induced changes in calcium homeostasis [30,32].

All differences between groups were observed at the time of repolarization of the inner mitochondrial membrane in the post-glutamate period. This may indicate that pathological processes triggered by mechanical injury affected the dynamics of the mitochondrial permeability, as well as the functionality of the electron transport chain.

## 3. Discussion

The pathological processes underlying the pathogenesis of TBI are schematically shown in Figure 9. A progressive increase in pathological phenomena after TBI is observed in 50% of the victims. The consequences of TBI include a number of neurological and psychiatric abnormalities that can be detected 2 to 10 years after injury in 90% of patients. These disorders usually determine the social disadaptation of patients, often leading to temporary disability, often turning into permanent disability [33]. In craniocerebral trauma, there are disorders of cerebral circulation, cerebrospinal fluid circulation, and permeability of the blood–brain barrier. Of course, the severity of pathological changes in the brain varies with different degrees of TBI severity [34]. Due to excessive water influx into brain cells and intercellular space, edema of the brain develops, which causes an increase in intracranial pressure. The processes of brain displacement and compression develop. This, in turn, causes further deterioration of blood circulation, metabolism, and functional activity of the brain. An adverse secondary factor of brain damage is its hypoxia due to respiratory or circulatory disturbances [33,35,36]. Thus, post-traumatic syndrome is a symptom complex that includes dizziness, headache, difficulties with concentration, nervousness, insomnia, and personality disorders.

There is a limited number of models that allow the assessment of the effect of mechanical trauma on Ca^2+^ homeostasis and the viability of primary cortical neurons [37]. Moreover, there are still no studies that have examined the acute (first seconds, minutes) and delayed (several days after mechanical injury) effects of mechanical trauma on cellular viability and the dynamic of changes in [Ca^2+^]_i_ and mitochondrial potential (ΔΨm) of neurons. The closest study is the one in which a stretch injury was applied to neurons and then their excitability was assessed using a patch clamp 30–60 min after the injury [38]. Another study also used a similar model of stretch injury, but in it, the viability and metabolic functions of neurons were recorded 24 h after the impact [39].

In the present work, we studied the healing dynamics of the mixed primary neuroglial culture under the conditions of in vitro experimental mechanical neurotrauma (Figure 1) and how it affects the pathological changes of calcium homeostasis and mitochondrial potential. The complexity of pathogenetic mechanisms mediating neurotrauma is explained by their high degree of heterogeneity and interconnectedness at both tissue and cellular levels (Figure 9). In an in vitro experimental model of neurotrauma, we have shown dysregulation of key homeostatic parameters ([Ca^2+^]_i_ and ΔΨm) of cells (Figure 6, Figure 7 and Figure 8), which may be responsible for the delayed death of neurons that survived the primary damage (Figure 2A). The negative effect of TBI on the ability of neurons to maintain calcium homeostasis under conditions of glutamate excitotoxicity is evident as early as 7 days after mechanical injury (Figure 7B). Increased calcium intake and the rapid onset of DCD in response to the subtoxic dose of glutamate are observed (Figure 8B_2_). In contrast, the recovery of initially low values of [Ca^2+^]_i_ is reduced (Figure 8D_2_).

These pathological processes may contribute to a decrease in cell survival in the areas located near the scratch. At the same time, it was shown that neurons and astrocytes migrate to the damaged area within two weeks after scratching (Figure 3 and Figure 4). Immunohistochemical staining for GFAP and β-III tubulin showed that the neurons migrate to the injured area earlier than other cells, but their repair potential is insufficient; neurons need neuroprotection at this time, as TBI reduces their ability to resist the pathological disturbances of calcium homeostasis.

Several parameters influence the magnitude of the integral fluorescence response during glutamate administration. The most important of them is the rate of the DCD onset, i.e., the multistep increase in [Ca^2+^]_i_, which is mirrored by the rapid increase in the Fura-FF fluorescent signal. The maximum fluorescence level in this phase of the experiment also depends on the efficiency of extra mitochondrial systems responsible for Ca^2+^ removal from the cytoplasm, such as membrane transporters PMCA, plasma NCX. The magnitude of the integral fluorescent response and the degree of [Ca^2+^]_i_ recovery during EGTA solution administration are additional parameters to assess the degree of Ca^2+^-homeostasis stability. Replacing the glutamate solution with EGTA solution not only reduces the deleterious activation of NMDA receptors activation, but also changes the Ca^2+^ ion concentration gradient between the cytoplasm and the external environment in the opposite direction, which allows cells that have not completely lost their homeostatic potential to remove Ca^2+^ from the cytoplasm. In Ca^2+^-free buffer, neurons from mechanically injured cultures poorly recovered the initial [Ca^2+^]_i_ compared to controls.

It is known that DCD is associated with permeabilization of the inner mitochondrial membrane. This phenomenon is accompanied by mitochondrial depolarization, which coincides in time with the increase in cytoplasmic Ca^2+^ concentration [30]. All differences between the groups were observed at the moment of repolarization of the inner mitochondrial membrane during the post-glutamate period. This may indicate that pathological processes triggered by mechanical injury affect the dynamics of the transition between the open and closed state of the mitochondrial permeability transition pore, as well as the functionality of the electron transport chain. It is known that [Ca^2+^]_i_ recovery can be promoted by the closure of mitochondrial permeability transition pores [40], which leads to the repolarization of the inner mitochondrial membrane and the recovery of the MCU providing the calcium return to the mitochondria.

We have shown that the significant [Ca^2+^]_i_ spike and synchronous depolarization of mitochondria could be caused by NMDA channel activation, as well as by other, yet unidentified mechanisms (Figure 6D,G). This conclusion is based on the fact that MK-801 blocked the high spike in [Ca^2+^]_i_ and the sharp drop in ΔΨm only in a part of the neurons (Figure 6E,H), whereas removing Ca^2+^ from the buffer abolished large changes in [Ca^2+^]_i_ and ΔΨm in all cells. In a mature neuroglial culture, the length of axons can reach hundreds of micrometers [41], while the length of astrocyte dendrites does not exceed 20–30 microns [42]. The relatively short distance from the scratch boundary (no more than ~100 µm), at which there was a strong increase in [Ca^2+^]_i_ and a significant drop in ΔΨm, suggests that some of the cells with such [Ca^2+^]_i_ and ΔΨm changes may be astrocytes. The small changes in [Ca^2+^]_i_ and ΔΨm at distances far from the scratch (≥100 μm) indicate that Ca^2+^ may enter neurons from the buffer at axon and/or dendrite cutting sites, diffusing through them and causing small changes in [Ca^2+^]_i_ in the soma. Of all the low-molecular-weight compounds, Glu and ATP are the most abundant in the cytosol (their concentration reaches 3–10 mM) [43]. Therefore, even after a significant (10–100 times) dilution of Glu and ATP when released from damaged cells in the scratch zone, they can stimulate the corresponding receptors on the surface of the neighboring undamaged cells.

Thus, mechanical injury causes strong and abrupt changes in [Ca^2+^]_i_ and ΔΨm in cells close to the damage zone. These changes are comparable with those caused by neurotoxic doses of glutamate. In some neurons, these changes are indeed caused by stimulation of Glu-directed NMDA receptors, whereas in other cells, ATP serves as the signal molecule. On the other hand, the maximal level of [Ca^2+^]_i_ in the glutamate washout phase depends on the efficiency of extra mitochondrial systems for Ca^2+^ removal from the cytoplasm, such as PMCA, and plasma NCX. Their efficiency is dependent directly or indirectly on the ATP concentration in the cytosol, which is determined by the intensity of cellular respiration. An increase in the [Ca^2+^]_i_ during exposure to glutamate in neurons from injured cultures compared to control neurons may indicate impaired ATP synthesis. Thus, mechanical trauma disrupts Ca^2+^ homeostasis systems in adjacent neurons. Impairment of calcium homeostasis in surviving neurons after TBI is not immediately apparent (Figure 7A,B) and may be the cause of post-traumatic syndrome. In this case, a protective therapy against [Ca^2+^]_i_ increase or ΔΨm loss under glutamate excitotoxicity may be promising.

## 4. Materials and Methods

### 4.1. Isolation and Cultivation of Cerebral Cortex Cells

Experiments with animals were performed in accordance with the ethical principles and regulatory documents recommended by the European Science Foundation (ESF) and the Declaration on Humane Treatment of Animals and in accordance with the Order of the Ministry of Health and Social Development of Russia no. 708n of 23 August 2010 “On Approval of Laboratory Practices.” Animal care, breeding, and experimental procedures were carried out as required of the Ethical committee of the Institute of General Pathology and Pathophysiology. Protocol no. 05-06/12 of 14.12.2017.

Wistar rat pups (P1–P2) were used for preparation of a primary mixed neuronal and glial culture of the cerebral cortex (primary neuroglial cultures) as previously described [18]. Briefly, the animals were anesthetized and decapitated, the brain was removed, and the cerebral cortex was isolated. A suspension of cortical neurons (10^6^ cells/mL) was obtained by treating brain tissue with papain (10 units/mL), dissociating by pipetting, washed from the cell debris by sedimentation in a centrifuge (200× *g*) in a Ca^2+^-free solution, then in Ca^2+^-containing solution and finally in the neurobasal medium (NBM, Gibco, Waltham, MA, USA). Cells were seeded in 48-well flat-bottom plates (2.5 × 10^5^ cells/well) (Costar, Glendale, AZ, USA) and in Petri dishes (ᴓ35 mm) with a glass insert (ᴓ14 mm) at the bottom (MatTeck, Ashland, MA, USA). Plates and Petri dishes were pre-coated with polyethyleneimine (0.05 mg/mL, 30 min). After one hour, 1.5 mL of NBM containing 2% Supplement B-27, 0.5 mM L-glutamine, and an antibiotics/antimycotic mixture (Gibco, Waltham, MA, USA) was added. The cells were incubated for 18 days in vitro (DIV) at 37 °C in an atmosphere of 5% CO_2_/95% air at 100% humidity.

### 4.2. Methods to Inflict Mechanical Injury

To measure changes in [Ca^2+^]_i_ and mitochondrial potential (ΔΨm) immediately after the trauma (the acute effect), we inflicted a single scratch trauma across the neuroglial culture with an insulin injection needle attached to the tip of a rod-like metal holder connected to a micromanipulator (Narishige Scientific Instrument Lab, Setagaya-ku, Tokyo 157-0062, Japan). In other experiments, the neuroglial cultures (4 DIV) were scratched across the diameter of the well bottom by a disposable sterile tip (Eppendorf, 200 µL). Then, we evaluated the delayed effect of mechanical damage on the cells viability and examined the long-term changes in [Ca^2+^]_i_ or ΔΨm on the 3rd (7 DIV) and 7th (11 DIV) days after scratching.

### 4.3. Evaluation of Neuronal Viability

Morphological analysis of the number of live and dead cells was performed as previously described [18]. Briefly, the cells were loaded with fluorescent dyes (15 min, 37 °C) using 1:1000 dilutions of DMSO stock solutions in a saline buffer. Syto-13 (1 μM, ex 485/em 530 nm) was employed to determine live cells. Necrotic cells were identified by staining with ethidium homodimer (EthD-1) (2 µM, ex 565/em 610 nm). The nuclei of both live and dead cells were stained with Hoechst 33342 (1 μM, ex 343/em 483). To obtain images (20× /NA = 0.45) and to count cells, the EVOS FL automated imaging system was employed. Neuronal viability was assessed using the ratio of living to dead cell numbers after 1 h, and at 3, 7, 10, and 14 days after injury (4, 7, 11, 14, and 18 DIV, respectively). The data obtained were normalized relative to the mean values in the control wells. The efficiency of injured cultures regeneration was defined as the number of neurons that moved to the damaged area.

### 4.4. Immunofluorescence Staining of Neuroglial Culture on Markers GFAP and Beta-3 Tubulin

The cell phenotype in the cultures was determined by immunofluorescence staining. The cultures were fixed with paraformaldehyde (4%, 15 min) and permeabilized with ice-cold methanol for 15 min. Nonspecific binding was blocked with 2% BSA in PBS (20 min, room temperature). The cells were incubated overnight at 4 °C with anti-β-III tubulin (2G10) mouse monoclonal antibodies (Thermofisher, #MA1-118; dilution of 1:100) and anti-GFAP chicken polyclonal antibodies (Thermofisher, #PA1-10004; dilution of 1:1000). Then, cells were washed with PBS and incubated (30 min, room temperature, dilution 1:100) with fluorescently labeled anti-mouse IgG (Thermofisher, #A32727; Alexa Fluor Plus 555) and anti-chicken IgY (Sigma Aldrich, #SAB4600031; CFTM 488A). For counting cells after immunofluorescent staining, their nuclei were labeled with Hoechst 33342 (1 μM, 10 min, room temperature).

Images of immunofluorescently labeled cells were obtained by employing an LSM 880 laser scanning confocal microscope equipped with an AiryScan module and GaAsP detector (Carl Zeiss, Jena, Germany). Tile scans (2 × 2; image size was 512 × 512 pixels) were collected with a Plan-Apochromat 40×/1.2 Imm Corr DIC M27 multi-immersion objective. Laser lines at 405 nm, 488 nm, and 561 nm were used to excite fluorescence of Hoechst 33342, and secondary anti-chicken IgY and anti-mouse IgG anti-bodies, respectively. Fluorescence emission was collected at 410–507 nm for Hoechst 33342, at 495–530 nm for anti-chicken IgY, and at 567–675 nm for anti-mouse IgG.

### 4.5. Fluorescence Microscopy Measurements of [Ca^2+^]_i_ and ΔΨm

Fluorescence measurements and results processing were performed as previously described [28] using a fluorescence imaging system, which consisted of an Olympus IX-71 inverted microscope equipped with a 175 W xenon lamp, 20 × fluorite objective, a Sutter Lambda 10-2 illumination system (Sutter Instruments, Novato, CA, USA), and a CoolSNAP HQ2 CCD camera operated by the computer program MetaFluor (Molecular Device, San Jose, CA, USA).

To measure changes in [Ca^2+^]_i_, cells were loaded with acetoxymethyl ester of low-affinity Ca^2+^ indicator Fura-FF (Fura-FF/AM). Stock solutions of Fura-FF/AM in dimethyl sulfoxide preliminary mixed with nonionic detergent Pluronic F-127 were added to cells in 1 mL of NBM to final concentrations 4 µM and 0.02%, respectively (60 min, 37 °C). For simultaneous monitoring of the [Ca^2+^]_i_ and ΔΨm changes, cells were loaded with a potential-sensitive dye rhodamine 123 (Rh123, 6.6 µM, 15 min, 37 °C) sensor. Fura-FF fluorescence was excited at 340 and 380 nm, and that of Rh123 at 485 nm. The accumulation of Rh123 in polarized mitochondria quenches the fluorescent signal. In response to depolarization, the fluorescence is dequenched [44]. The fluorescence emission of both dyes was monitored at 525 ± 15 nm. The time-lapse of the Fura-FF signals’ recording was 3 s for the first 3 min after the mechanical injury application, and then the time-lapse was switched to 30 s to avoid the Ca^2+^ indicator photo-bleaching.

The measurements were performed in a “normal buffer” (NB) containing (mM): 130 NaCl, 5.4 KCl, 2 CaCl_2_, 1 MgCl_2_, 20 HEPES, 5 D-glucose; pH 7.4 was adjusted with 1 M NaOH. Glutamate (Glu, 33 µM) was added in Mg^2+^-free NB-containing glycine (3.3 µM). Nominally calcium-free NB contained 2 mM MgCl_2_ and 0.1 mM EGTA instead of Ca^2+^. The maximal Fura-FF signals were determined using the Ca^2+^ ionophore ionomycin (2 µM) in the presence of 5 mM CaCl_2_ (without Mg^2+^) at the very end of the experiments.

The amount of Ca^2+^ stored in mitochondria was estimated by depolarizing the organelles with the protonophore carbonyl cyanide 4-(trifluoromethoxy) phenylhydrazone (FCCP, 1 µM) in nominally Ca^2+^-free NB. All solutions were prepared on the day of the experiments.

### 4.6. Reagents

The cell culture supplies were obtained from Invitrogen (Thermo Fisher Scientific, Waltham, MA, USA). Fluorescence microscopy dyes were acquired from Molecular Probes (Thermo Fisher Scientific, Waltham, MA, USA). All other reagents were purchased from Sigma-Aldrich (Merck, St. Louis, MO, USA). We also used several conventional media, optimal for the particular types of experiments.

### 4.7. Statistical Analysis

Statistical analysis was performed using GraphPad Prism 6 (GraphPad Software Inc., San Diego, CA, USA). At least three experiments were carried out in each series. The data were checked for normality using three tests: D’Agostino-Pearson omnibus normality test, Shapiro–Wilk normality test, and KS normality test. The data are presented according to statistical rules depending on the type of data distribution. The Kruskal–Wallis test with Dunn’s multiple comparisons correction, the Mann–Whitney test, or the t-test was used to analyze the [Ca^2+^]_i_ and ΔΨm kinetics (described in detail in the figure captions).

## 5. Conclusions

We developed a model of mechanical neurotrauma that allows us to measure the dynamics of [Ca^2+^]_i_ and ΔΨm both in the first seconds, minutes, and several days after injury. NMDA receptor ion channels serve as the main pathway for Ca^2+^ entry into neurons in the event of mechanical damage to a neuronal culture. Ca^2+^ overload of mitochondria and a prolonged decrease in ΔΨm can cause delayed neuronal death and thus play a key role in the development of the post-traumatic syndrome. We were also able to visualize the process of the neuroglial culture repair, where neurons were the first in the damaged area and were followed by astrocytes. Nevertheless, the viability of neurons inevitably decreased two weeks after injury. This effect may be attributed to the long-term metabolic dysregulation of neurons caused by trauma. Therefore, the use of neuroprotective compounds would allow us to test this hypothesis.

## Figures and Tables

**Figure 1 ijms-23-03858-f001:**
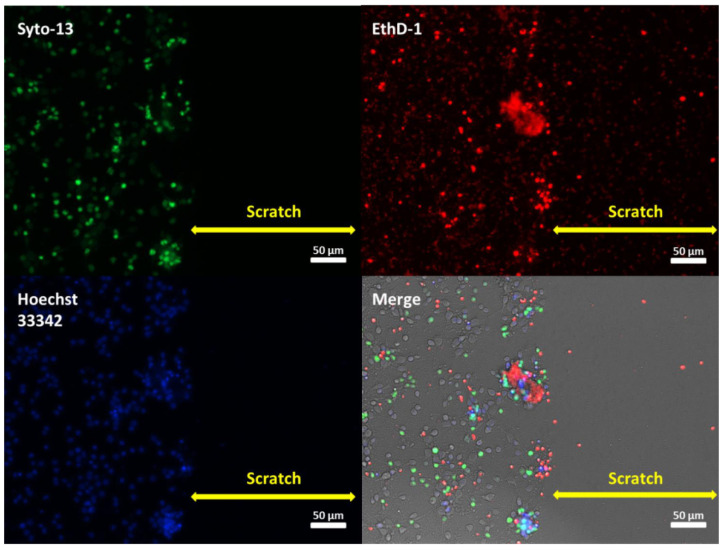
Fluorescence images of the neuroglial culture with injury (7 DIV). Cell culture (4 DIV) was mechanically damaged and stained with fluorescent dyes after 3 days. Live cells were stained with Syto-13; dead cells were stained with EthD-1; nuclei were stained with Hoechst 33342. Fluorescent images were superimposed on transmitted light images (Merge). Scale bar is shown on the images. Fluorescence images show a change in the proportion of live and necrotic cells in cell culture: the edges of the scratch are even; no cells or nuclei remain in the damage area. The merged image clearly shows a cluster of dead cells along the border of the lesion; there are no live cells in the scratch zone itself; only a few dead cells are visible.

**Figure 2 ijms-23-03858-f002:**
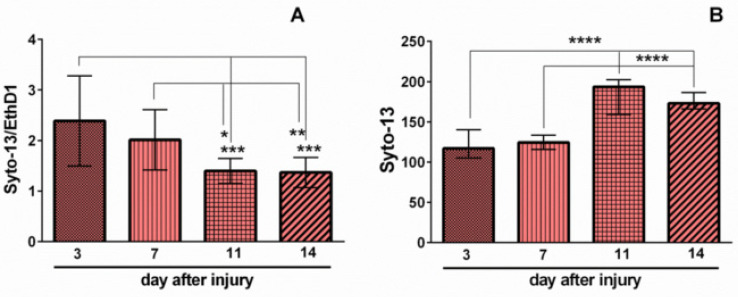
Analysis of the cell survival and regenerative potential of the injured neuroglial cell culture on days 3, 7, 11, and 14 (7, 10, 15, and 18 DIV, respectively). (**A**) Changes in the proportion of live and necrotic cells of the primary neuroglial culture in nearby areas at a distance no further than 250 µm from the edge of the scratch. (**B**) Number of live cells in the scratch area. Data are presented as mean ± SD. ***** *p* < 0.05; ** *p* < 0.01; *** *p* < 0.005; ******** *p* < 0.001 (conventional one-way ANOVA, with Tukey’s multiple comparisons test).

**Figure 3 ijms-23-03858-f003:**
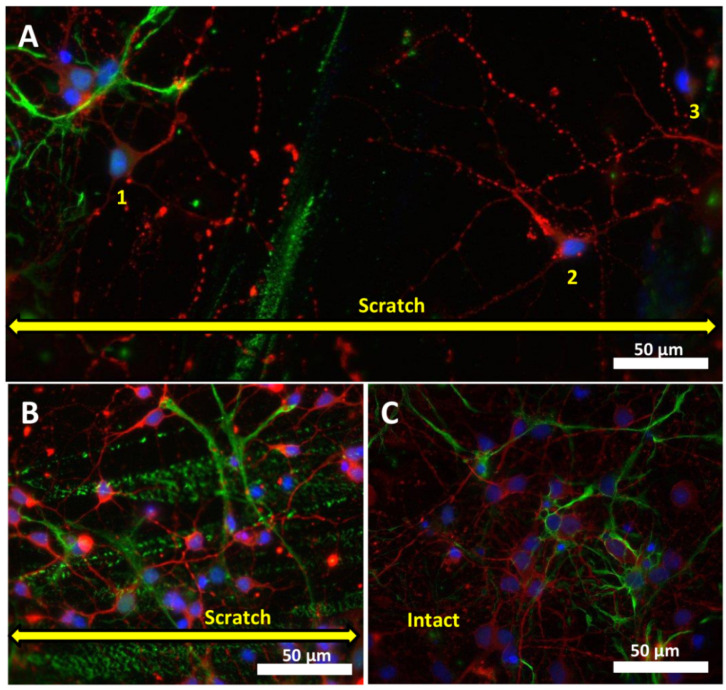
Fluorescence staining of neuroglial culture for cell markers GFAP and β-III tubulin. Fluorescent images of cell culture on days 3 (**A**) and 7 (**B**) after scratching (7 DIV and 10 DIV, respectively) and intact cell culture (10 DIV) without scratching (**C**) were taken with a fluorescent microscope Zeiss Axiovert-200, objective 40×/NA = 1.35 (oil). Scale bar is shown on the images. The yellow double arrow corresponds to the width of the scratch. Pairwise stitching (**A**) of two adjacent fragments was obtained using ImageJ software. (**A**) On day 3 after injury, a few neurons are observed in the injured area (indicated by numbers 1–3). (**B**) Numerous neurons migrate into the damaged area on day 7 after scratching.

**Figure 4 ijms-23-03858-f004:**
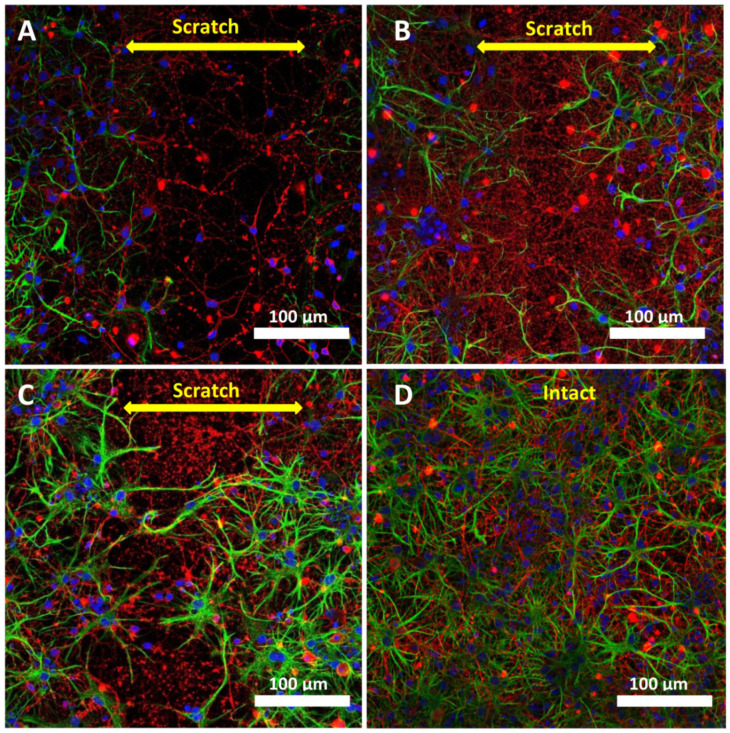
Fluorescence staining of neuroglial culture for GFAP и β-III tubulin. Fluorescent images of the cell culture were obtained on day 3 after scratching, 7 DIV (**A**); on day 7 after scratching, 10 DIV (**B**); on day 10 after scratching, 14 DIV (**C**), and intact cell culture without scratching, 14 DIV (**D**). Scale bar is shown on the images. The yellow double arrow corresponds to the width of the scratch. Images were acquired using a LSM 880 scanning laser confocal microscope equipped with an AiryScan module and GaAsP detector. The ZEN Black tile scanning function was used to stitch four separate images in each panel.

**Figure 5 ijms-23-03858-f005:**
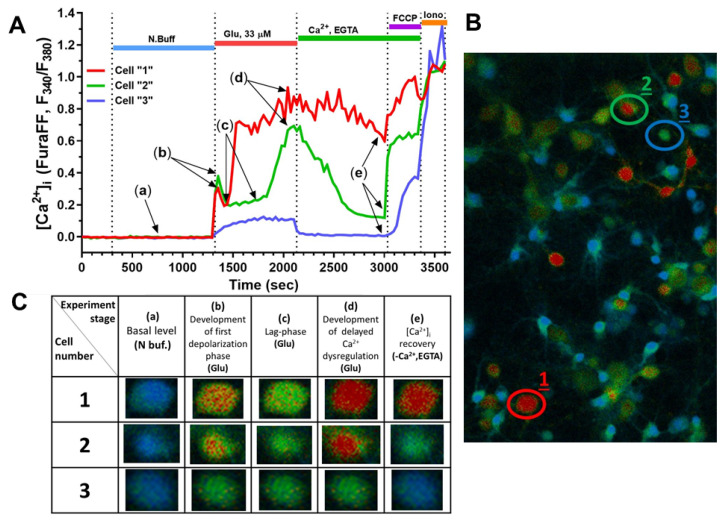
Different profiles of the dynamics in the ratiometric signal of Ca^2+^-sensitive probe Fura-FF. (**A**) Dynamic curves of [Ca^2+^]_i_ changes in three neurons (1, 2, 3) marked in the image in panel (**B**). (**B**) Fluorescent microscopy image of primary neuroglial culture during exposure to Glu. (**C**) Images of individual neurons marked in Photo (**B**), reflecting different levels of [Ca^2+^]_i_ at different stages of the experiment. In cell 1, DCD develops; cell 2 develops DCD with [Ca^2+^]_i_ recovery; and no DCD is observed in cell 3.

**Figure 6 ijms-23-03858-f006:**
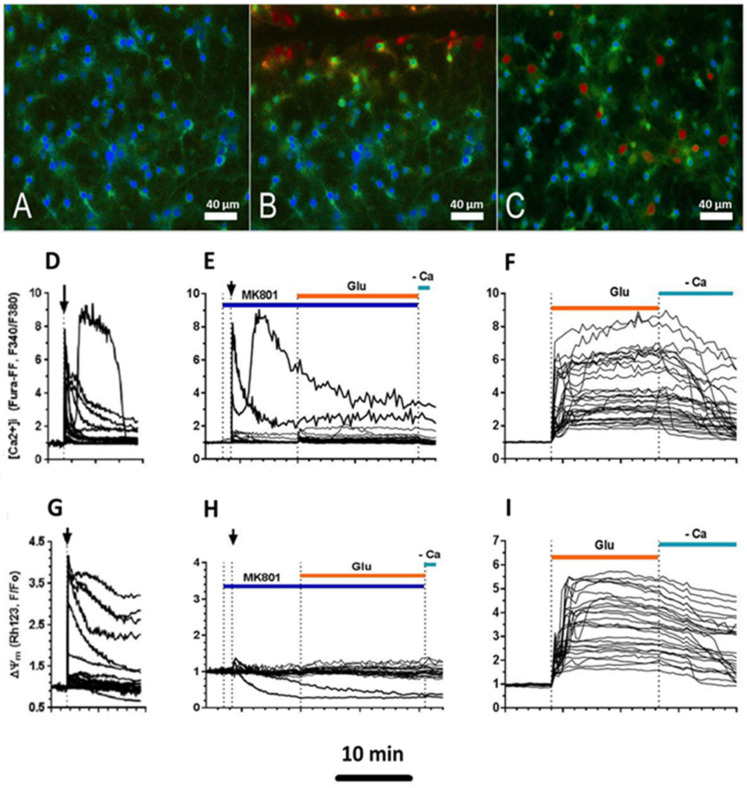
Fluorescent microscopy images of the primary culture of rat cortical cells of the rat before scratching (**A**), 10 s after scratching (**B**), and intact culture 60 s after the addition of glutamate (**C**). The cultures were loaded with low-affinity Ca^2+^ indicator Fura-FF. A higher concentration of intracellular Ca^2+^ corresponds to a warmer color in the image. Glu (100 μM) was added in a nonmagnesium buffer in the presence of Gly (10 μM). Scale bar is shown on the images. Changes in Ca^2+^ concentration (**D**–**F**) and mitochondrial potential (ΔΨm) (**G**–**I**) induced by scratching (**D**,**E**,**G**,**H**) and Glu addition (**F**,**I**). To determine the contribution of ionotropic glutamate receptors to changes in [Ca^2+^]_i_ and ΔΨm, the NMDA inhibitor MK-801 (10 μM) was added to the culture (**E**,**H**). Changes in [Ca^2+^]_i_ are presented as ratios of fluorescent Ca^2+^ indicator Fura-FF signals measured at 340 and 380 nm excitation (F340/F380) and recorded at 525 nm. In each neuron, the F340/F380 ratio is normalized to the baseline value at rest. Changes in ΔΨm are represented as changes in the signal of the potential-sensitive probe Rh123 (**G**–**I**) normalized to the value in resting cells (F/Fo). An increase in the Rh123 signal corresponds to a decrease in ΔΨm. The vertical arrows on panels (**D**,**E**,**G**,**H**) indicate the moment of scratching.

**Figure 7 ijms-23-03858-f007:**
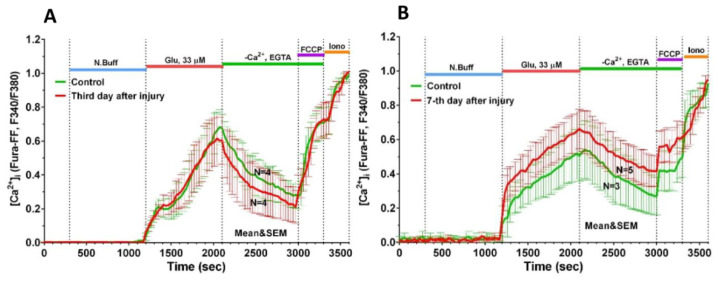
Averaged dynamics of Fura-FF fluorescent signal changes during the experiment (Glu 33 µM (**A**) on day 3 (7 DIV) and day 7 (11 DIV) (**B**) after scratching. Data are presented as mean ± SEM; N is the number of experiments per group. Cells were incubated with Glu solution for 15 min to create glutamate excitotoxicity during TBI. The Glu solution was then replaced with nominally calcium-free EGTA solution to alleviate Ca^2+^ extrusion from the cytoplasm. After 5 min of FCCP incubation, the solution was replaced and the cells were exposed to ionomycin, which makes the membrane permeable to Ca^2+^ and thus allows estimation of the maximum cytoplasmic capacity for calcium.

**Figure 8 ijms-23-03858-f008:**
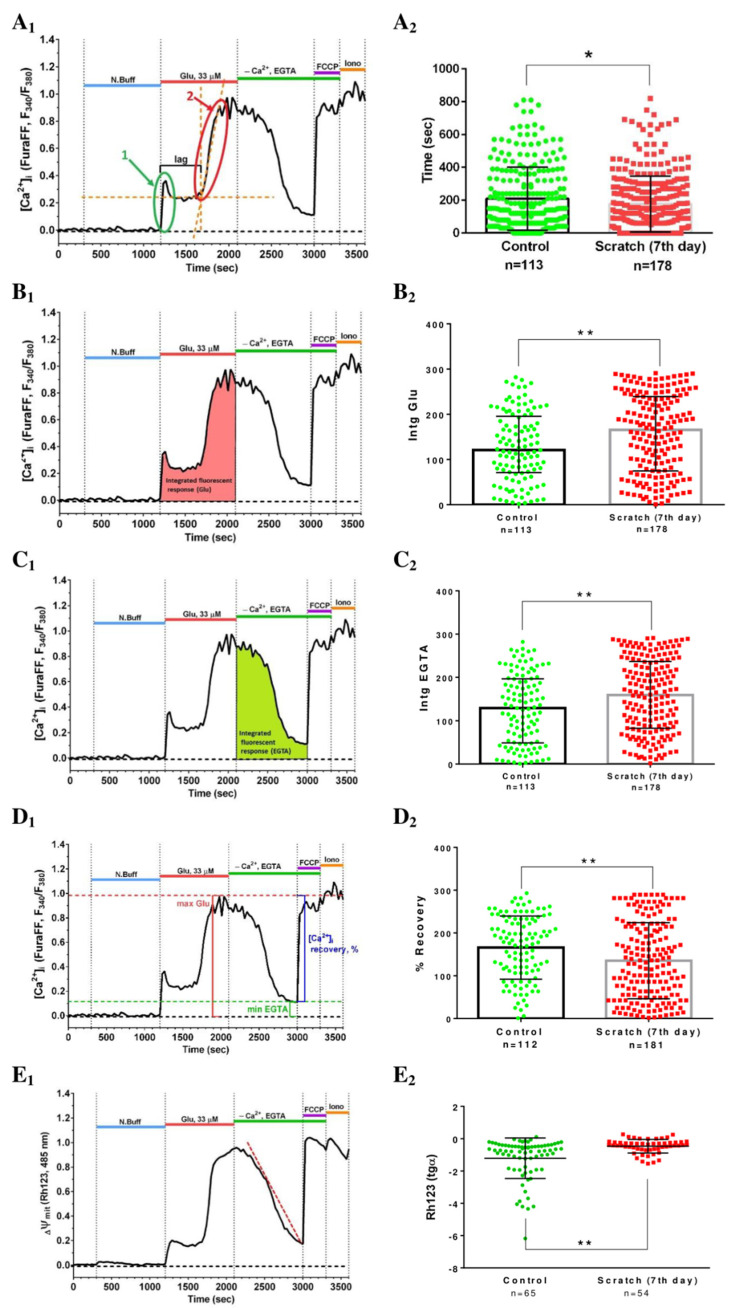
The parameters of the dynamic changes in calcium homeostasis during the experiment. (**A**–**E**)_1_—Parameter measurement methods. (**A**–**E**)_2_—Differences in parameters between the studied groups. ***** *p* < 0.05; ****** *p* < 0.01. (**A**)—Lag-phase before the onset of delayed calcium dysregulation (DCD). First and second phase of [Ca^2+^]_i_-rise, respectively, are indicated by arrows 1 and 2 to evaluate the lag-period of DCD. (**B**)—Integrated fluorescent response (Glu). (**C**)—Integrated fluorescent response (EGTA). (**D**)—Recovery of [Ca^2+^]_i_ to the baseline values during application of EGTA solution. (**E**)—Linear approximation (red dotted line) of the signal curve of the potential-sensitive fluorescent dye Rh123 during application of EGTA solution The slopes (tg α) of linear approximations in panel E_1_ are depicted for control and injured cultures in panel E_2_ (title of Y-axis).

**Figure 9 ijms-23-03858-f009:**
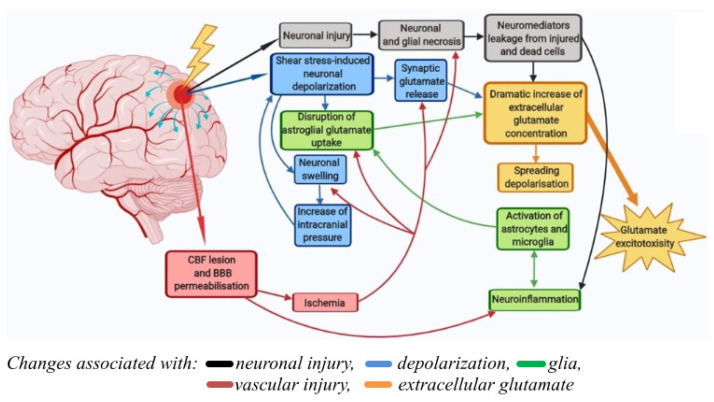
Pathological processes underlying the pathogenesis of traumatic brain injury. Primary damage at TBI is due to the mechanical impact itself (force, degree of penetration, and area of damage) and develops in the first few seconds after the trauma (black arrows). Secondary damage is longer-term (hours-days) and is mediated by primary damage mechanisms. They include pathological depolarization due to shear stress (blue arrows), changes in CBF dynamics, as well as neuro-inflammatory processes caused by glial and immune cells coming from the bloodstream during BBB permeabilization (green and red arrows). The delayed neuronal death caused by TBI is due to the excitotoxicity of glutamate (yellow arrow).

## Data Availability

Not applicable.

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
