# Peer review of "Acute and Delayed Effects of Mechanical Injury on Calcium Homeostasis and Mitochondrial Potential of Primary Neuroglial Cell Culture: Potential Causal Contributions to Post-Traumatic Syndrome"

_ijms, 2022, doi:10.3390/ijms23073858_

Round 1

Reviewer 1 Report

This manuscript reports data on acute and delayed cell viability and the changes of Ca2+ and mitochondrial potential in neuroglial cell cultures following mechanical scratch injury as an in vitro model of traumatic brain injury. The main new data concern the Ca2+ dynamics immediately after injury and upon glutamate application or protection from Glu excitotoxicity, as well as the simultaneous determination of changes in mitochondrial potential. Ca2+ dynamics and mitochondrial potential changes are also determined in the cultures after various delay times. Given earlier studies in similar systems the results are not unexpected and add relatively limited new information. Detailed interpretation and mechanistic conclusions are hampered by a lack of single-cell identification and molecular precision, hence much additional work will be needed in the future to dissect the underlying mechanisms.

A general comment: It is not clear how the method of in vitro mechanical trauma used in this study differs significantly from previously used systems - the difficulty of standardized parameters for mechanical trauma dimensions and evaluation of trauma severity are not (and can hardly be) addressed in this study.

Specific points:

  1. Title: The title implies that the authors’ findings have revealed the cause of TBI. This is obviously an overstatement. The last part of the title after the semicolon (“the cause of post-traumatic syndrome”) should be modified, e.g., to “potential causal contributions to post-traumatic syndrome”.
  2. Figure 1 legend: “cores” stained with Hoechst… Do you mean “nuclei stained with Hoechst…?
  3. Results, line 101/102 and Fig. 2B: It is not true that the RATIO of live to dead cells increased 11 and 14 days after injury: Fig. 2A shows a lower ratio (~1.5) of Syto-13 over EthD-1 cells at 11 and 14 days; Fig. 2B shows a higher number of Syto-13 cells at these times but the lower ratio of live/dead cells implies that the number of EthD-1 cells must be even greater at these times. The other problem with this Figure is that it is not clear where exactly the cells were counted (area and distance from the scratch edge).
  4. Figure 3A: What is meant by “pairwise stitching composite”? It seems that the magnification in 3A is higher (twice?) that in 3B and 3C - if so, the scale bars in 3A and 3B/C can not be identical.
  5. Figure 6A-C: Scale bar is not shown.
  6. Results, lines 195-200: Fig. 7 is erroneously mentioned here instead of Fig. 6.
  7. Fig. 8A1: What is indicated by arrows 1 and 2 in Fig. 8A1? In the legend, the statistical meaning of the single or double asterisks should be indicated.
  8. Methods 4.3 and 4.4 and Fig. 2: How was quantification performed? How many fields of cells were counted and in what area (e.g.,  0.2 x 0.2 mm) and how far from the edge of the scratch?
  9. Grammar/language should be carefully checked and corrected, e.g., on line 304 “excessive watering of brain cells and intercellular space” probably means “excessive water influx into brain cells and intercellular space”. Parts of the Discussion are difficult to understand and/or grammatically incorrect. For example, on lines 335/6 “Perhaps, these pathological processes…”; lines 355-58 “In spite of the fact…”; lines 370-72 “We have shown that significant jump-like growth…”
  10. Refs. 16, 25, and 34 are incomplete. 

Author Response

Author's Reply to the Review Report (Reviewer 1):

This manuscript reports data on acute and delayed cell viability and the changes of Ca2+ and mitochondrial potential in neuroglial cell cultures following mechanical scratch injury as an in vitro model of traumatic brain injury. The main new data concern the Ca2+ dynamics immediately after injury and upon glutamate application or protection from Glu excitotoxicity, as well as the simultaneous determination of changes in mitochondrial potential. Ca2+ dynamics and mitochondrial potential changes are also determined in the cultures after various delay times. Given earlier studies in similar systems the results are not unexpected and add relatively limited new information. Detailed interpretation and mechanistic conclusions are hampered by a lack of single-cell identification and molecular precision, hence much additional work will be needed in the future to dissect the underlying mechanisms.

A general comment: It is not clear how the method of in vitro mechanical trauma used in this study differs significantly from previously used systems - the difficulty of standardized parameters for mechanical trauma dimensions and evaluation of trauma severity are not (and can hardly be) addressed in this study.

We are grateful to Reviewer-1  for a thorough analysis of the work and for comments that made it possible to improve the text of the manuscript!

Indeed, we used not a unique, but a simple and most accessible method for all researchers to mechanically damage a cell culture. The novelty of this study is a combined approach to the study of the phenomena associated with neurotrauma in vitro. We observed the immediate response of cells to injury in the acute period (seconds) and the delayed effects of neurotrauma by measuring the same parameters several days after injury. Previously, there were no studies on the simultaneous measurement of [Ca2+]i and ΔΨm during the first seconds after injury. The integral fluorescent response of Rh123 was found to have a similar trend of differences as for Fura-FF (Figure 6 E and G), which confirms the involvement of mitochondria in mechanical damage-induced changes in calcium homeostasis. In the second series of experiments, when studying the delayed effects of scratching, significant differences between groups were observed during repolarization of the mitochondrial membrane in the postglutamate period. This fact may indicate a change in mitochondrial permeability due to disruption of the electron transport chain. The study of changes in calcium homeostasis is associated with a complex analysis. We propose for analysis those parameters that are worth paying attention to (Figure 8 A1-E1), taking into account the close relationship in the work of mitochondria and transport systems associated with calcium pumping. In our work, we obtained significant differences between the control and «scratch» groups (Figure 8 A2-E2). In future studies, this will allow us to have a clear idea of the neuroprotective effects of the tested drugs. We also plan to continue experiments to study the underlying mechanisms of neurotrauma by adding molecular precision. Thank you for pointing that out!

Specific points:

Point 1: Title: The title implies that the authors’ findings have revealed the cause of TBI. This is obviously an overstatement. The last part of the title after the semicolon (“the cause of post-traumatic syndrome”) should be modified, e.g., to “potential causal contributions to post-traumatic syndrome”.

Response 1: Thanks for your comment and valuable advice! The obtained results cannot fully explain the cause of the post-traumatic syndrome. The last part of the title after the semicolon (“the cause of post-traumatic syndrome”) was modified to your example “potential causal contributions to post-traumatic syndrome”. Line 4

Point 2: Figure 1 legend: “cores” stained with Hoechst… Do you mean “nuclei stained with Hoechst…?

Response 2: The error in the Figure 1 legend has been fixed…"nuclei" instead of "cores". Line 113; in corrected version Line 122

Point 3: Results, line 101/102 and Fig. 2B: It is not true that the RATIO of live to dead cells increased 11 and 14 days after injury: Fig. 2A shows a lower ratio (~1.5) of Syto-13 over EthD-1 cells at 11 and 14 days; Fig. 2B shows a higher number of Syto-13 cells at these times but the lower ratio of live/dead cells implies that the number of EthD-1 cells must be even greater at these times. The other problem with this Figure is that it is not clear where exactly the cells were counted (area and distance from the scratch edge).

Response 3: The corresponding edit to Figure 2B was introduced to the text «…the number of living cells increased..» instead of «…the ratio of live to dead cells increased…» in line 101/102; in corrected version lines 112/114. We propose that the living cells appear in the scratch area via migration from the undamaged parts of the culture. This assumption is based on a fact that there are no living cells in the damaged area directly after the scratch (Fig. 1).The sentence corresponding to Figure 2A in lines 98/99; in corrected version Lines 106/107 describes the decrease in the life/dead cell ratio in area located in direct vicinity from the “scratch”. Living and the dead cells were counted not farther then 250 µm from the edge of damaged area. In the legend of the Figure 2A has been added «… in nearby areas at a distance no further than 250 μm from the edge of the scratch» in corrected version Lines 137/138.

Point 4: Fig. 3A: What is meant by “pairwise stitching composite”? It seems that the magnification in 3A is higher (twice?) that in 3B and 3C - if so, the scale bars in 3A and 3B/C can not be identical.

Response 4: The error in the Figure 3 legend has been fixed…"pairwise stitching" instead of "pairwise stitching composite". The program ImageJ stitches the borders of two photos using the function “pairwise stitching ”. Due of this, we able visualized the whole neuron with all its neurites. The magnification (40x) for Figure 3A-C is the same, as well as the scale bars.

Point 5: Fig. 6A-C: Scale bar is not shown.

Response 5: Scale bar has been added in the Figure 6A-C.

Point 6: Results, lines 195-200: Fig. 7 is erroneously mentioned here instead of Fig. 6.

Response 6: The error has been fixed … «Figure 6 B, D» instead of " «Figure 6B, Figure 7D» in line 195 in corrected version Line 206. The error has been fixed … «Figure 6C, F and I» instead of " «Figure 6C and Figure 7F, I» in line 200; in corrected version Line 211.

Point 7: Fig. 8A1: What is indicated by arrows 1 and 2 in Fig. 8A1? In the legend, the statistical meaning of the single or double asterisks should be indicated.

Response 7: First and second phase of [Ca2+]i -rise respectively was indicated by arrows 1 and 2 in to evaluate the lag-period of DCD. This clarification has been added in the legend Figure 8A1. We indicated the statistical meaning of the single or double asterisks in the legend: *p < 0.05 or **p < 0.01 respectively in Figure 8A2-E2. Line 270.

Point 8: Methods 4.3 and 4.4 and Fig. 2: How was quantification performed? How many fields of cells were counted and in what area (e.g.,  0.2 x 0.2 mm) and how far from the edge of the scratch?

Response 8: The width of the damaged area was 600±10 µm, which was evaluated using 4x magnification. Detailed visualization of life and dead cells we used 20x magnification. Half of the microscope field of view was taken by the area of scratch, another half – by the undamaged area. Therefore, the square of each field (damaged or undamaged) was 90000±2000 µm2 at images. In each well, 6 photographs were taken along the scratch. Living and the dead cells were counted not farther then 250 µm from the edge of damaged area. We used 48-well flat-bottom plates. There were 6 experiments in total.

Point 9: Grammar/language should be carefully checked and corrected, e.g., on line 304 “excessive watering of brain cells and intercellular space” probably means “excessive water influx into brain cells and intercellular space”. Parts of the Discussion are difficult to understand and/or grammatically incorrect. For example, on lines 335/6 “Perhaps, these pathological processes…”; lines 355-58 “In spite of the fact…”; lines 370-72 “We have shown that significant jump-like growth…”

Response 9: All the grammar/language has been checked and corrected by English editing editorial office of the «BIOCHEMISTRY (MOSCOW) SUPPLEMENT. SERIES A: MEMBRANE AND CELL BIOLOGY». 

Point 10: Refs. 16, 25, and 34 are incomplete. 

Response 10: Refs. 16, 25, and 34 have been completed.

All fixes are shown in the pdf-file. Please see the attachment.

Reviewer 2 Report

The authors presented a simple and easily reproducible model of mechanical neurotrauma for in vitro studies of primary and secondary neuronal damage. This model allows to study the acute (sec, min) and delayed (7 days after the mechanical impact) influence of scratch injury on cellular viability, dynamic of [Ca2+]i and mitochondrial potential (ΔΨm) changes in neurons. By studying these parameters, they followed the dynamics of the scratch damage healing and determined which neuroglial culture cells were the first to migrate to the mechanical injury zone. Thus, they have validated a new in vitro model of TBI, which could further be used for studying neuroprotective compounds previously tested in the experimental model of glutamate excitotoxicity.

The manuscript was well written and the conclusion was supported by the experimental data. I have some minor concerns.

  1. Regarding the edema and increased intracellular sodium concentration after TBI should be discussed.
  2. Typo errors were found in lines 37 and 3335 regarding the living and dead cells.
  3. Line 296, in this sentence, TBI should be used instead of its full name.

Author Response

Author's Reply to the Review Report (Reviewer 2)

The authors presented a simple and easily reproducible model of mechanical neurotrauma for in vitro studies of primary and secondary neuronal damage. This model allows to study the acute (sec, min) and delayed (7 days after the mechanical impact) influence of scratch injury on cellular viability, dynamic of [Ca2+]i and mitochondrial potential (ΔΨm) changes in neurons. By studying these parameters, they followed the dynamics of the scratch damage healing and determined which neuroglial culture cells were the first to migrate to the mechanical injury zone. Thus, they have validated a new in vitro model of TBI, which could further be used for studying neuroprotective compounds previously tested in the experimental model of glutamate excitotoxicity.

The manuscript was well written and the conclusion was supported by the experimental data. I have some minor concerns.

We are grateful to Reviewer-2  for a thorough analysis of the work and for comments that made it possible to improve the text of the manuscript!

Point 1: Regarding the edema and increased intracellular sodium concentration after TBI should be discussed.

Response 1: Thanks for your valuable comment! We wanted to focus our study on changing [Ca2+]i. However, we added We added a paragraph regarding the edema and increased intracellular sodium concentration after TBI into the introduction chapter (lines 74/83 in corrected version ). In our study we did not measure neither the alteration of intracellular sodium concentration upon trauma or induced glutamate excitotoxicity, nor the parameters of edema which followed them.

Point 2: Typo errors were found in lines 37 and 3335 regarding the living and dead cells.

Response 2: Thanks for pointing that out! It was required «live/dead cells» instead of «living/dead cells». But, the sentence was removed in accordance with the Reviewer-1 comments.

Point 3: Line 296, in this sentence, TBI should be used instead of its full name.

Response 3: The error has been fixed in the line 296… «TBI» instead of «traumatic brain injury» in corrected version line 308

Round 2

Reviewer 1 Report

The authors have answered all questions and adequately revised their manuscript. However, they could have included their answer concerning Methods 4.3./4.4 related to the cell counting in the text (under Materials and Methods).